# Diversity of RNA Viruses and Circular Viroid-like Elements in *Heterobasidion* spp. in Near-Natural Forests of Bosnia and Herzegovina

**DOI:** 10.3390/v17081144

**Published:** 2025-08-20

**Authors:** László Benedek Dálya, Ondřej Hejna, Marcos de la Peña, Zoran Stanivuković, Tomáš Kudláček, Leticia Botella

**Affiliations:** 1Department of Forest Protection and Wildlife Management, Faculty of Forestry and Wood Technology, Mendel University in Brno, Zemědělská 3, 613 00 Brno, Czech Republic; xdalya@mendelu.cz (L.B.D.); tomas.kudlacek@mendelu.cz (T.K.); 2Department of Genetics and Biotechnologies, Faculty of Agriculture and Technology, University of South Bohemia in České Budějovice, Na Sádkách 1780, 370 05 České Budějovice, Czech Republic; hejna@fzt.jcu.cz; 3Instituto de Biología Molecular y Celular de Plantas, Universidad Politécnica de Valencia-CSIC, 46022 Valencia, Spain; delapenya@gmail.com; 4Department of Integrated Protection of Forest Ecosystems, Faculty of Forestry, University of Banja Luka, Vojvode Stepe Stepanovića Blvd. 75a, 78 000 Banja Luka, Bosnia and Herzegovina; zoran.stanivukovic@sf.unibl.org; 5Department of Forest Ecology, Faculty of Forestry and Wood Technology, Mendel University in Brno, Zemědělská 1, 613 00 Brno, Czech Republic

**Keywords:** diversity, forest virome, mycoviruses, viroids, *Heterobasidion annosum*, *Heterobasidion abietinum*, root rot

## Abstract

*Heterobasidion* root rot fungi represent a major threat to conifer forest stands, and virocontrol (biocontrol) has been proposed as an alternative strategy of disease management in recent years. Here, we investigated the occurrence of RNA viruses and viroid-like genomes in *Heterobasidion annosum* sensu lato in near-natural forests of Bosnia and Herzegovina (Dinaric Alps), a region previously unexplored in this regard. Seventeen *H. annosum* s.l. isolates were screened for virus presence by RNA Sequencing and bioinformatic analyses. In total, 32 distinct mycoviruses were discovered in the datasets, 26 of which were previously unknown. The detected viruses represent two dsRNA (*Partitiviridae* and *Curvulaviridae*), six linear ssRNA (*Mitoviridae*, *Narnaviridae*, *Botourmiaviridae*, *Virgaviridae*, *Benyviridae*, and *Deltaflexiviridae*) and three circular ssRNA (*Dumbiviridae*, *Quambiviridae*, and *Trimbiviridae*) virus families. In addition to the known circular ambiviruses with their hammerhead (HHRz) and hairpin (HPRz) ribozymes, two other smaller non-coding circular RNAs of ca. 910 bp each were identified encoding HHRz and deltavirus (DVRz) ribozymes in both polarities of their genomes. This study documents the first report of a putative viroid-like RNA agent in *Heterobasidion*, along with beny-like and deltaflexivirus-like viruses in *Heterobasidion abietinum*, and expands the known virosphere of *Heterobasidion* species in Southeastern European forests.

## 1. Introduction

Bosnia and Herzegovina has approximately 2.9 million ha of forests, which is about 57% of its total area [1]. The tree species composition is characterized by the dominance of oak (33%), pure beech (33%) and mixed coniferous–broadleaved stands (24%). In addition to providing significant commercial benefits [1], Bosnian forests are considered as a biodiversity hotspot giving shelter to numerous endemic and relict species [2]. The old-growth forests of the Dinaric Alps are especially important from naturalistic, cultural, and scientific perspectives not only regionally but also on the continental scale [3]. The importance of these ecosystems is increasing because of their role as a reference for the assessment of climate change impact in European forests [4]. Only small areas of valuable mixed stands of Norway spruce (*Picea abies*), silver fir (*Abies alba*), and European beech (*Fagus sylvatica*) remain intact to this day. Such stands are situated in the Janj, Lom, and Perućica old-growth forest reserves in the Republic of Srpska [5] and their vegetation belongs to the *Piceo-Abieti-Fagetum dinaricum* forest association [3].

Conifer tree species in Bosnia seem more vulnerable to climate change-induced environmental stress than broadleaves [6,7], and are therefore prone to higher degrees of biotic and abiotic damage [8]. The most important rotting fungal species contributing to the spruce declining process in Bosnia are *Heterobasidion parviporum* and *Armillaria ostoyae* [9,10]. *Heterobasidion abietinum* is also known to cause root and heart rot, which often leads to the drying out of fir trees in the territory of the Republic of Srpska [11]. The effectiveness of currently available forest protection methods to control *Heterobasidion annosum* sensu lato is insufficient. In recent years, the use of mycoviruses as biocontrol agents (BCAs) has been proposed as an alternative strategy against the pathogen, and a field trial in Finland [12] showed the hypovirulence-inducing potential of Heterobasidion partitivirus 13 from *H. annosum* (HetPV13-an1).

Comprehensive studies in Sweden [13], Finland [14], and Czechia [15] revealed the diverse virome of *Heterobasidion* spp. These investigations typically involved local, i.e., boreal or Central European *Heterobasidion* isolates, while some included specimens from other geographic regions. At present, little is known about viral communities hosted by Southeast European *Heterobasidion* strains. Greek *H. annosum* s.l. strains have been screened for dsRNA presence [16,17], but no further virus screenings were performed on material from the Balkans. The present work attempted to reduce this knowledge gap by exploring the occurrence and phylogenetic relationships of viruses associated with *H. annosum* s.l. in Bosnia.

## 2. Materials and Methods

### 2.1. Collection of Heterobasidion Isolates

Fruiting bodies of *H. annosum* s.l. were collected from coniferous forests of Bosnia, primarily from near-natural stands. Axenic cultures were isolated from basidiocarps and cultivated on 2% malt extract agar (MEA—HiMedia, Mumbai, Maharashtra, India) at 18 °C. In total, 17 *Heterobasidion* strains were included in the study (Figure 1 and Appendix A), and were identified to the species level as described previously [15], following the dilution protocol of the Phire^®^ Plant Direct PCR Kit (Thermo Fisher Scientific, Waltham, MA, USA).

### 2.2. RNA Extraction

*Heterobasidion* cultures were grown on cellophane-covered MEA plates at 22 °C in the dark for 2 weeks. After grinding the mycelia in liquid N_2_ using a sterile mortar and pestle, total RNA was extracted using RNAzol^®^ RT (Sigma-Aldrich, Steinheim, Germany). Homogenization was performed by shaking for 1 min at 30 Hz in a Mixer Mill MM 400 (Retsch, Haan, Germany). RNA quality was visually assessed using a 1.2% agarose gel. RNA concentrations were measured by a Qubit^®^ 2.0 Fluorometer (Thermo Fisher Scientific, Waltham, MA, USA). The RNA solutions were stored at −80 °C.

### 2.3. Total RNA Sequencing

Three pools consisting of equal volumes with 0.5–1.5 µg of total RNA of six *H. abietinum* (NGS dataset ‘ABI’), three *H. parviporum* (‘PAR’), and seven *H. annosum* s.l. (six *H. abietinum* and one *H. annosum*; ‘BIH’) strains, as well as total RNA of a single *H. annosum* strain (‘2120’; Appendix A), were sent for sequencing to the Institute of Applied Biotechnologies a.s. (Olomouc, Czechia). Ribosomal RNA was removed using the NEBNext^®^ rRNA Depletion Kit v2 (New England Biolabs, Ipswich, MA, USA). The cDNA libraries were constructed using the NEBNext^®^ Ultra II Directional RNA Library Prep Kit for Illumina^®^ (New England Biolabs, Ipswich, MA, US) and sequenced in paired-end (PE) (2 × 151 bp) on a NovaSeq 6000 (Illumina Inc., San Diego, CA, USA).

### 2.4. Bioinformatics

Each of the NGS datasets consisted of 588–955 million PE reads. Basecalling, adapter clipping, and quality filtering were performed using Bcl2fastq v2.20.0.422 Conversion Software (Illumina Inc., San Diego, CA, USA). The quality of raw reads was assessed using FastQC v0.11.9 [18]. Trimming of N bases, adapter sequences, and low-quality ends (<30) were processed with Cutadapt v4.7 [19]. Reads <50 bp were discarded from subsequent analyses. Post-trimming quality was reassessed with FastQC, and overrepresented ribosomal RNA sequences from *Heterobasidion* were removed using SortMeRNA v4.3.6 [20], with the SILVA database of rRNA serving as a reference (https://www.arb-silva.de/). STAR v2.7.9a was employed with default settings for read mapping to the host genome [21]. The NCBI GenBank assembly GCA_001457955.1 was used as the reference genome for *H. annosum*. After alignment, all mapped reads were discarded, and only unmapped reads were retained. To validate the presence of known viruses in the remaining reads, BWA v0.7.17-r1188 [22] was used to map the reads to NCBI viral RefSeq sequences at https://www.ncbi.nlm.nih.gov/labs/virus/vssi/#/ (1 February 2025). The coverage was calculated for each reference virus sequence by SAMtools v1.16.1 [23]. Coverage plots of every viral contigs were visualized in Geneious Prime^®^ 2025.2.1 and they are available in 10.6084/m9.figshare.29634521.

Unmapped reads were then used for de novo assembly using SPAdes v3.15.5 with default settings [24]. Contigs shorter than 500 bp were discarded after this step. Final contigs were cross-referenced with several databases using BLAST v2.10.0 [25]. As a reference, we used a list of viral reference sequences from NCBI. BLASTn was applied for the nucleotide database accessible at https://www.ncbi.nlm.nih.gov/labs/virus/vssi/#/virus?SeqType_s=Nucleotide. Similarly, BLASTx was employed for the protein database available at https://www.ncbi.nlm.nih.gov/labs/virus/vssi/#/virus?SeqType_s=Protein, and also to search the UniProt database for a specified virus taxon at https://www.uniprot.org/uniprot/?query=taxonomy:10239. Contigs with an e-value higher than 1 × 10^−3^ were discarded. For BLASTn, the nt database from NCBI, available at https://ftp.ncbi.nlm.nih.gov/blast/db was used. For BLASTx, both the nr database from NCBI, accessible at https://ftp.ncbi.nlm.nih.gov/blast/db and UniProt Knowledgebase (UniProtKB), available at https://www.uniprot.org/uniprotkb/, were employed. The BLAST search was restricted to the best hit per contig. Within each RNA-Seq dataset, the virus candidate contigs (i.e., longer than 500 bp) were imported to Geneious Prime^®^ 2024.0 (https://www.geneious.com) and subjected to further examination. After the manual correction of misassembles, the resulting viral genome sequences were saved and used in further analyses. The search for viral sequences encoding open reading frame (ORF) regions which lack significant similarity to known genes in databases (so-called ORFan analysis) was conducted as described in [26].

The screening of small self-cleaving ribozymes was carried out using the INFERNAL software v. 1.1.5. [27], which identifies RNA elements using secondary structure and nucleotide covariation modeling. Covariance models for the small ribozymes recently described from diverse ambiviruses, mitoviruses, and other viroid-like agents were used for the searches [28]. Secondary RNA structures of minimum free energy of the ambivirus genomes were calculated with the RNAfold program from the ViennaRNA Package 2.0 [29] and visualized by circular plots obtained with the jupiter software (https://github.com/rcedgar/jupiter).

### 2.5. Phylogenetic Analysis and Pairwise Comparison

Maximum Likelihood phylogenetic trees based on the amino acid (aa) sequences of either the RdRP or the replication protein were constructed using the bootstrapping algorithm implemented in MEGA11 [30]. The model with the lowest Bayesian Information Criterion score was selected for each tree.

Pairwise (pw) comparisons of viral sequences were performed in Geneious Prime^®^ version 2025.2.1 using the MAFFT alignment algorithm.

### 2.6. DAPC Analysis of Ambiviruses

The dataset for DAPC analysis included 43 ambivirus RdRP nucleotide sequences from Finland (13 ambiviruses), Czechia (16 ambiviruses), and Bosnia (14 ambiviruses). DAPC analysis was performed using the package [31,32]. The number of axes retained in the Principal Component Analysis (PCA) step was set to 30. The number of axes retained in the Discriminant Analysis step was set to the number of populations minus one.

## 3. Results and Discussion

### 3.1. Virus and Viroid-like Agents Discovery

In total, 47 mycoviral and 2 mycoviroid-like sequence segments were reconstructed from contigs assembled using the RNA-Seq data covering 12 *H. abietinum*, 3 *H. parviporum*, and 2 *H. annosum* isolates (Table 1 and Appendix A). Some of these segments were concatenated from two or three contigs. Interestingly, the negative-sense antigenome was assembled for 36% of the detected viral segments. No viral contigs were identified in *H. annosum* isolate 2120. One variant of Heterobasidion ourmia-like virus 1 (HetOlV1; *Magoulivirus alphaheterobasidion*) [33] was found in dataset PAR. The rest of the viruses were detected in datasets ABI and BIH (Table 1). The discovered segments belong to 3 distinct dsRNA viruses classified into two families and 29 ssRNA viruses representing nine families (Table 1). Six of these viruses were previously known. For eight viruses, two or three distinct variants were identified in either the same or different datasets (Table 1).

Additionally, seven ORFan contigs, 600–2300 nucleotides (nt) in length, were identified in datasets ABI and PAR (Appendix A). These may indicate the presence of putative viral segments with previously unknown genome organization.

### 3.2. Alphapartitiviruses

Four contigs with similarities to partitiviruses were found, either to an RdRP domain (50–70% identity with the first hit in BLASTx) or a coat protein (CP; 43–50%; Table 1). The contigs were deemed to represent two bisegmented partitiviruses named HetPV32 and HetPV33, which were paired based on 20 identical nt at the 5’ end of two of them (Figure 2). Conservation of the first 7–45 nt in the 5′ UTR of the sense RNA between the two genome segments is common in *Partitiviridae*. A stretch of identical nt at the beginning of the 5′ UTR was also observed in HetPV8 [34] and HetPV21 [33]. The length of the RdRP encoding dsRNA1 segment of HetPV32 and HetPV33 was ca. 2 kb, and the length of their coat or capsid protein (CP) encoding dsRNA2 segment was 1.8 kb (Figure 2), which falls within the typical range for alphapartitiviruses. The dsRNA1 segments of HetPV32 and HetPV33 had polyA tails of 18 and 32 bp, respectively. The two novel partitiviruses shared 40% pairwise (pw) aa identity in their RdRP and merely 22% pw aa identity in the CP. Both viruses were hosted by *H. abietinum* (Table 1), which was also the natural host of one HetPV1 variant from Greece [16]. Interestingly, the read coverage of HetPV32 dsRNA1 was an order of magnitude higher than that of dsRNA2, while for HetPV33, the difference was five-fold in the opposite direction (Table 1). This observation may reflect differences in the accumulation levels of the two genome segments in the host mycelium. Unique ratios of RdRP and CP genome segments of *Heterobasidion* partitiviruses were previously found [35]. Phylogenetic analysis of the *Durnavirales* order placed the novel partitiviruses in the *Alphapartitivirus* genus (Figure 3). HetPV33 is very closely related to HetPV4 and HetPV5, while HetPV32 fell into a branch separate from other alphapartitiviruses of *Heterobasidion*.

**Table 1 viruses-17-01144-t001:** Viruses and viroid-like elements detected in the present study.

Mycovirus	Acronym	Segment	GenBank Accession	Host Species	Length (nt)	% GC	Mapped Reads ^f^	Mean Depth	BLASTX First ^g^	Identity	Query Cover	E Value
Heterobasidion partitivirus 32	HetPV32	dsRNA1	PP626369	* H. abietinum *	1934 ^d^	47.7% ^d^	260,018 ^d^	18,931 ^d^	Sarcosphaera coronaria partitivirus (QLC36816)	50.1%	89%	0
		dsRNA2	PP626370	* H. abietinum *	1816	54.7%	25,327	1939	Carrot cryptic virus (ACL93279)	43.0%	70%	3.0 × 10^−92^
Heterobasidion partitivirus 33	HetPV33	dsRNA1	PP626371	* H. abietinum *	1970 ^d^	46.5%^d^	12,617 ^d^	916 ^d^	Heterobasidion partitivirus 4 (ADV15443)	70.5%	94%	0
		dsRNA2	PP626372	* H. abietinum *	1755	53.3%	59,967	4803	Tulasnella partitivirus 3 (BDB07480)	49.9%	83%	1.0 × 10^−130^
Heterobasidion RNA virus 6	HetRV6	dsRNA1	PP626373	* H. abietinum * OR *H. annosum*	1996	56.3%	1172	78	Heterobasidion RNA virus 6 (ADW82833)	93.1%	91%	0
		dsRNA2	PP626374	* H. abietinum * OR *H. annosum*	1748	63.4%	915	69	Heterobasidion RNA virus 6 (QED55787)	93.7%	62%	0
Heterobasidion mitovirus 4	HetMV4	RNA1	PP626375	* H. abietinum * ^ a ^	5292	42.1%	67,004	1685	Heterobasidion mitovirus 3 (QED55406) ^h^	69.2% ^h^	46% ^h^	0 ^h^
Heterobasidion mitovirus 5	HetMV5	RNA1	PP626376	* H. abietinum *	3715	40.9%	676,330	25,276	Heterobasidion mitovirus 1 (AIF33766) ^h^	63.6% ^h^	71% ^h^	0 ^h^
Heterobasidion mitovirus 6	HetMV6	RNA1	PP626377	* H. abietinum *	3521	41.4%	644,309	26,115	Heterobasidion mitovirus 1 (AIF33766) ^h^	63.4% ^h^	74%^h^	0 ^h^
Heterobasidion mitovirus 7	HetMV7	RNA1	PP626378	* H. abietinum * OR *H. annosum* ^b^	4314	41.1%	538,636	17,624	Lentinula edodes mitovirus 1 (QOX06058)^h^	38.8% ^h^	51%^h^	2.0 × 10^−160h^
Heterobasidion narna-like virus 1	HetNlV1	RNA1	PP626379	* H. abietinum * ^ a ^	3956	54.0%	34,130	1221	Heterobasidion narna-like virus 1 (UHK02569)	92.8%	93%	0
		RNA2	PP626380	*H. abietinum* ^a^	3998	55.8%	27,091	960	Heterobasidion narna-like virus 1 (UHK02570)	91.8%	92%	0
Heterobasidion narna-like virus 5	HetNlV5	RNA1	PP626381	* H. abietinum * OR *H. annosum*	3953	55.3%	77,698	2714	Heterobasidion narna-like virus 4 (WBE16499)	78.9%	96%	0
Heterobasidion narna-like virus 6	HetNlV6	RNA1	PP626382	* H. abietinum * OR *H. annosum*	1731 ^e^	51.2	4745	379	Mbeech associated narna-like virus 5 (WPR16644)	53.0%	93%	0
		RNA2	PP626383	* H. abietinum * OR *H. annosum*	2252	51.5	6832	394	Botrytis cinerea binarnavirus 2 (QLF49184)	45.9%	92%	0
Heterobasidion ourmia-like virus 1	HetOlV1	RNA1	PP626384	* H. parviporum *	2601	49.5	214,903	11,422	Heterobasidion ourmia-like virus 1 (UHK02571)	96.6%	73%	0
Heterobasidion ourmia-like virus 2	HetOlV2	RNA1	PP626385	*H. abietinum* ^a^	2169 ^e^	58.0	10,251	640	Heterobasidion ourmia-like virus 2 (UOX39320)	93.0%	67%	0
Heterobasidion ourmia-like virus 6	HetOlV6	RNA1	PP626386	* H. abietinum *	2516	51.2	12,337	658	Heterobasidion ourmia-like virus 1 (WLV75612)	76.9%	74%	0
Heterobasidion ourmia-like virus 7	HetOlV7	RNA1	PP626387	*H. abietinum* ^a^	2366	50.5	11,364	566	Heterobasidion ourmia-like virus 1 (WLV75612)	79.4%	72%	0
Heterobasidion ourmia-like virus 8	HetOlV8	RNA1	PP626388	* H. abietinum *	2305	50.4	16,611	884	Heterobasidion ourmia-like virus 1 (WLV75612)	77.1%	76%	0
Heterobasidion ourmia-like virus 9	HetOlV9	RNA1	PP626389	* H. abietinum * OR *H. annosum*	2505	50.3	118,718	6411	Heterobasidion ourmia-like virus 1 (WLV75612)	81.4%	74%	0
Heterobasidion tobamo-like virus 2	HetTlV2	RNA1	PP626390	* H. abietinum * OR *H. annosum*	6675 ^e^	49.3	1880	38	Heterobasidion tobamo-like virus 1 (UOX39322)	87.6%	64%	0
Heterobasidion beny-like virus 1	HetBlV1	RNA1	PP626391	*H. abietinum* OR *H. annosum* ^a^	7563 ^d^	57.2 ^d^	268,134 ^d^	4956 ^d^	Rhizoctonia cerealis beny-like virus (WMI40034)	31.1%	64%	2.0 × 10^−162^
Heterobasidion deltaflexivirus 1	HetDFV1	RNA1	PP626392	*H. abietinum* OR *H. annosum* ^a^	6928	64.1	123,445	2322	Rhizoctonia solani flexi-like virus 1 (QDW81317)	33.1%	34%	2.0 × 10^−107^
Heterobasidion ambi-like virus 1	HetAlV1	RNA1	PP744446	* H. abietinum *	4754	49.2	822,963	24,597	Heterobasidion ambi-like virus 1 (UHK02572)	97.8%	38%	0
Heterobasidion ambi-like virus 2	HetAlV2	RNA1	PP744447	* H. abietinum * OR *H. annosum*	4286	49.2	938,195	30,286	Heterobasidion ambi-like virus 2 (UHK02574)	95.6%	41%	0
Heterobasidion ambi-like virus 30	HetAlV30	RNA1	PP744448	* H. abietinum *	5085	49.2	8663	241	Heterobasidion ambi-like virus 6 (UOX39308)	58.0%	42%	0
Heterobasidion ambi-like virus 31	HetAlV31	RNA1	PP744449	*H. abietinum* ^c^	4876	49.9	32,263	922	Heterobasidion ambi-like virus 3 (UHK02576)	68.3%	42%	0
Heterobasidion ambi-like virus 32	HetAlV32	RNA1	PP744450	* H. abietinum *	4450	49.2	1,172,645	37,056	Heterobasidion ambi-like virus 1 (UHK02572)	83.3%	40%	0
Heterobasidion ambi-like virus 33	HetAlV33	RNA1	PP744451	* H. abietinum *	4368	49.2	505,254	16,380	Heterobasidion ambi-like virus 2 (UHK02574)	92.6%	40%	0
Heterobasidion ambi-like virus 34	HetAlV34	RNA1	PP744452	* H. abietinum * OR *H. annosum*	4956	50.2	110,813	3101	Heterobasidion ambi-like virus 3 (UHK02576)	67.3%	42%	0
Heterobasidion ambi-like virus 35	HetAlV35	RNA1	PP744453	* H. abietinum * OR *H. annosum*	4851	48.4	11,912	336	Heterobasidion ambi-like virus 8 (UOX39312)	61.6%	40%	0
Heterobasidion ambi-like virus 36	HetAlV36	RNA1	PP744454	* H. abietinum * OR *H. annosum*	4660	49.7	61,156	1779	Heterobasidion ambi-like virus 12 (WOK44145)	76.0%	44%	0
Heterobasidion ambi-like virus 37	HetAlV37	RNA1	PP744455	* H. abietinum * OR *H. annosum*	4767	48.8	54,368	1574	Heterobasidion ambi-like virus 8 (UOX39312)	65.9%	41%	0
Heterobasidion ambi-like virus 38	HetAlV38	RNA1	PP744456	* H. abietinum * OR *H. annosum*	4762	48.0	87,791	2543	Heterobasidion ambi-like virus 8 (UOX39312)	66.3%	41%	0
Heterobasidion ambi-like virus 39	HetAlV39	RNA1	PP744457	* H. abietinum * OR *H. annosum*	4632	49.0	21,324	602	Heterobasidion ambi-like virus 10 (WOK44143)	74.1%	44%	0
Heterobasidion ambi-like virus 40	HetAlV40	RNA1	PP744458	* H. abietinum * OR *H. annosum*	4287	48.4	53,587	1644	Heterobasidion ambi-like virus 12 (WOK44145)	87.7%	45%	0
Heterobasidion circular RNA 1	HetcRNA1	RNA1	PQ298351	* H. abietinum *	919	60.1	206,319	33,675	n.s.			
Heterobasidion circular RNA 2	HetcRNA2	RNA1	PQ298352	* H. abietinum *	918	60.3	22,977	3754	n.s.			

^a^ Another virus variant was hosted by *H. abietinum* or *H. annosum*. ^b^ Two other virus variants were hosted by *H. abietinum*. ^c^ Another virus variant was hosted by *H. abietinum*. ^d^ Without polyA tail. ^e^ Partial genome sequence. ^f^ Using Geneious assembler with medium-low sensitivity. Corresponding BAM files with mapped reads for each viral contig are available at 10.6084/m9.figshare.29634521. ^g^ Search performed on 31.07.2024. ^h^ Using the Mold Protozoan Mitochondrial genetic code. n.s., no significant similarity found.

### 3.3. Orthocurvulavirus Annosi

A new variant of Heterobasidion RNA virus 6 (HetRV6; *Orthocurvulavirus annosi*) was detected, sharing 93–94% BLASTx identity with the variant HetRV6-ab6 ([17]; Table 1). The 2 kb long dsRNA1 segment corresponded to the RdRP, while dsRNA2, encoding a hypothetical protein (HP) with unknown function, was 1.7 kb long (Figure 2). A stretch of 39 identical bases was noticed at the 5′ termini of both segments (Figure 2). Comparable levels of conservation at the same region were documented in HetRV6-ab6 and -pa36 (GenBank OR644495, OR644496), as well as in orthocurvulaviruses of *Curvularia protuberata* [36], *Sclerotium hydrophilum* [37], *Trichoderma harzianum* [38], and *Lactarius rufus* [39]. Both segments of the new HetRV6 variant had a low depth of coverage compared to most other viruses reported here (Table 1), in agreement with [33]. Phylogenetically, HetRV6 formed a highly supported branch with Phlebiopsis gigantea curvulavirus 1 [40] within the *Curvulaviridae* family (Figure 3).

**Figure 3 viruses-17-01144-f003:**
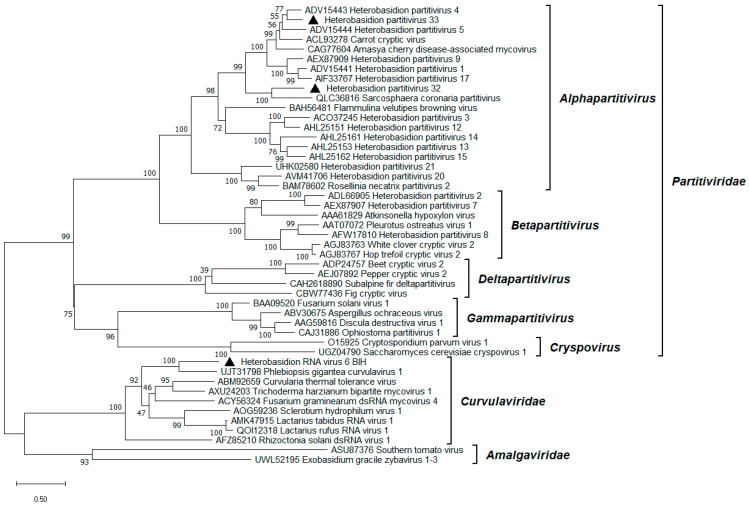
Phylogenetic relationships within and between relevant families of the *Durnavirales* order. The *Amalgaviridae* family served as outgroup. The tree was built based on the alignment of RdRP aa sequences generated with MUSCLE in MEGA11 [30]. The evolutionary history was inferred by using the Maximum Likelihood method and the Le_Gascuel_2008 model [41]. A discrete Gamma distribution was used to model evolutionary rate differences among sites (five categories +G +I). All positions with less than 95% site coverage were eliminated; i.e., fewer than 5% alignment gaps, missing data, and ambiguous bases were allowed at any position. Evolutionary analyses were conducted in MEGA11 with 1000 bootstrap repeats. The percentage of trees in which the associated taxa clustered together is shown next to the branches. Branch lengths are proportional to the number of substitutions per site. Alphapartitiviruses and the orthocurvulavirus variant described in this study are denoted by a triangle.

### 3.4. Mitoviruses

Four novel mitoviruses, named Heterobasidion mitovirus 4–7 (HetMV4–7), were derived from our data (Table 1). Their genomes were 3.5–5.3 kb in length and contained the mitoviral RdRP conserved domain (Figure 2). HetMV7 also possessed a short downstream ORF encoding an HP (Figure 2). HetMV4 earned the title of the longest mitovirus described to date ahead of HetMV3-an1 [42]. HetMV4 was represented by two virus variants, while HetMV7 by three (Table 1), with each variant sharing 98% aa pw identity over their RdRP. *Heterobasidion abietinum* was recorded as a host of each mitovirus described here (Table 1). The read coverage values of HetMV5–7 were among the highest observed in the present study (Table 1). The high accumulation levels of mitoviruses in *Heterobasidion* and other plant pathogenic fungi are a common phenomenon [15]. HetMV7 clustered with Lentinula edodes mitovirus 1 [43], while HetMV4–6 with HetMV1 [44] and HetMV3 [42] within the *Duamitovirus* genus of the *Mitoviridae* family (Figure 4).

### 3.5. Narna-like Viruses

Two new variants of Heterobasidion narna-like virus 1 (HetNlV1) [33] were detected, one of which surely inhabited *H. abietinum*. Each segment of both variants showed over 92% identity in BLASTx (Table 1). Another contig, almost 4 kb long and sharing 79% aa identity in its predicted RdRP with HetNlV4 [15], was designated as HetNlV5 RNA1 segment (Table 1; Figure 2). Two shorter sequences (~1.7 and 2.3 kb) were considered to constitute the genome of a novel bisegmented narna-like virus named HetNlV6 (Table 1; Figure 2). Their most significant BLASTx hits were Mbeech associated narna-like virus 5 and Botrytis cinerea binarnavirus 2 with 53% and 46% identity, respectively. Due to being incomplete at the 5’ end of its RNA1 segment and lacking the GDD motif within the RdRP, HetNlV6 was excluded from the phylogenetic analysis. All the other *Heterobasidion* narna-like viruses clustered tightly within *Narnaviridae* (Figure 4).

### 3.6. Ourmia-like Viruses

A new variant of HetOlV1 with 97% BLASTx identity to variant HetOlV1-RKU3.2.124 [33] was found in *H. parviporum* (Table 1), which remains the only host reported for this virus. Further, two variants of HetOlV2, each sharing 93% identity in BLASTx with the variant HetOlV2-CZ2072 [15], were revealed (Table 1). Both were incomplete at the 3’ end, and one variant infected *H. abietinum* (Table 1; Figure 2). Another four distinct ourmia-like viruses with high similarity (77–81% BLASTx identity) to HetOlV1 were discovered and named HetOlV6–9 (Table 1). Their monosegmented genomes of 2.3–2.5 kb contain a single predicted ORF encoding for a putative RdRP (Figure 2). HetOlV6–8 were present in *H. abietinum* (Table 1). HetOlV7 was represented by two different variants (Table 1). Pw aa identities of the RdRP between HetOlV2 and HetOlV7 variants were 98% and 97%, respectively. Phylogenetic analysis of the *Lenarviricota* phylum indicated that *Heterobasidion* ourmia-like viruses most likely belong to the *Magoulivirus* genus within the family *Botourmiaviridae* (Figure 4).

### 3.7. Tobamo-like Virus

A ~6.7 kb long sequence sharing 88% BLASTx identity with the recently described Heterobasidion tobamo-like virus 1 (HetTlV1) [15] was named HetTlV2 (Table 1). Only a partial genome of HetTlV2 could be reconstructed from our data. Based on pw alignments with the HetTlV1 genome sequence, there was a gap of ca. 773 bp between the truncated RdRP encoding ORF and the movement protein (MP) encoding ORF of the HetTlV2 genome (Figure 2). Furthermore, the HP at the 3’ end was incomplete. Most likely, HetTlV2 caused a low titer infection, manifested by the lowest read coverage in the present study, with a mean depth of 38 reads (Table 1). Oddly, the full genome of HetTlV1 was reconstructed at a mean depth of just 20 reads [15]. Therefore, RNA-Seq of a single host isolate may be a viable approach to obtain the complete genome sequence [15]. As opposed to HetTlV1, the first ORF of HetTlV2 encodes an HP. The largest ORF was predicted to encode a replicase comprising methyltransferase and helicase domains. The downstream ORF encodes the RdRP. Similarly to HetTlV1, −1 ribosomal frameshifting is presumably utilized to express these two proteins. The ORF encoding an MP contained the DEAD-like helicase superfamily and the helicase conserved C-terminal domains (Figure 2). *Heterobasidion* tobamo-like viruses occupied a separate phylogenetic branch and seem to be distantly related to other members of the family *Virgaviridae* (Figure 5).

### 3.8. Beny-like Virus

A 7.6 kb long contig showing the highest similarity (31% BLASTx identity) to Rhizoctonia cerealis beny-like virus was designated as Heterobasidion beny-like virus 1 (HetBlV1; Table 1). Its first ORF, containing the DEXS-box helicase and *Benyviridae* RdRP conserved domains, was projected to encode an RdRP (Figure 2). A shorter ORF encoding an HP was located downstream. The genomic sequence includes a polyA tail of 12 nt. HetBlV1 and another HetBlV1 variant detected in the same dataset (Table 1) shared 93% and 91% pw aa identities in their RdRP and HP, respectively. Classified members of the *Benyviridae* family infect plants, while the extension of this taxonomic group seems imminent as numerous viruses with similar properties have been recently discovered in fungal or insect metagenomic studies [46]. Phylogenetic analysis of the *Alsuviricetes* class placed HetBlV1 into a well-supported cluster consisting of mycoviruses and an entomovirus within *Benyviridae* (Figure 6).

### 3.9. Deltaflexivirus

As another novelty in the virome of *Heterobasidion* spp., we report the discovery of Heterobasidion deltaflexivirus 1 (HetDFV1), represented by a 6.9 kb long contig, with Rhizoctonia solani flexi-like virus 1 as its most significant BLASTx hit (33% identity; Table 1). The HetDFV1 genome encompassed a single predicted ORF encoding an RdRP, which included methyltransferase, helicase, and *Deltaflexiviridae* RdRP domains (Figure 2). Two HetDFV1 variants were present in the same dataset (Table 1) with 78% identity in their RdRP aa sequence. The *Deltaflexiviridae* family has traditionally accommodated plant and fungal viruses, although several affiliated viruses have been found in animals by metagenomics in recent years [47]. According to phylogenetic analysis, deltaflexiviruses could be of monophyletic origin, as no signs of cluster formation were apparent among putative members of the family (Figure 6).

### 3.10. Ambi-like Viruses

One contig detected in dataset ABI showed 98% BLASTx identity to the virus variant Heterobasidion ambi-like virus 1 (HetAlV1; *Orthodumbivirus unoheterobasidii*)-4R1 known from *H. parviporum* ([33]; Table 1). Another contig shared 96% BLASTx identity with the virus variant HetAlV2 (*Orthodumbivirus duoheterobasidii*)-RKU3.1.25 ([33]; Table 1). Both of these new variants had robust read support (Table 1). Eleven new viruses with moderate to high similarity to previously described ambi-like viruses of *Heterobasidion* were discovered and designated as HetAlV30–40 (Table 1). Each virus had a genome of 4.3–5.1 kb, containing two bidirectional ORFs. In the case of HetAlV30, an incomplete dimer of the genome sequence was assembled, which is common due to the circular genome organization of *Ambiviricota* [15,48,49,50]. The majority of the ambi-like viruses possessed hammerhead (HHRz) or hairpin (HPRz) ribozymes positioned at the C-termini of one or both ORFs (Appendix A). The genome organization of HetAlV31, representative of the novel ambi-like viruses, is shown in Figure 2. HetAlV30–33 dwelled in *H. abietinum* and HetAlV31 had two variants (Table 1) with 90% nt pw identity in the RdRP. The ambi-like viruses of *Heterobasidion* were placed in three different families, where they grouped tightly together (Figure 7). Of the newly discovered ambiviruses, HetAlV30 belongs to the *Quambiviridae* family, HetAlV32 and HetAlV33 to *Dumbiviridae*, while the majority fell into *Trimbiviridae*.

The dataset BIH yielded 37 additional short contigs resembling ambi-like viruses of *Heterobasidion* (Appendix A). These fragments have either zero or a single complete ORF, encoding either the putative RdRP or the HP. As we could not reliably compile them into coding complete genomes, we cannot confirm whether they belong to the viruses described above or to different ones.

The DAPC analysis of the ambiviruses’ genetic differentiation based on their host variant geographical origin (Finland, Czechia, and Bosnia) shows clear country-wise separated clusters (Figure 8, Appendix A) and low overlap between them. Bosnia and Czechia are more separated along Discriminant Function 1 and Finland separated primarily along Discriminant Function 2. The tight clustering of Bosnian and Czech ambiviruses indicates low within-group variation, while the broader spread of Finnish samples suggests higher heterogeneity. These results suggest that the genetic variation of *Heterobasidion* ambiviruses is strongly influenced by geographic origin, potentially reflecting local evolutionary dynamics or host–virus associations. This pattern is particularly common among mycoviruses due to their intracellular transmission mode [51].

### 3.11. Viroid-like cRNAs

Two small circular RNAs (cRNAs) of 918 and 919 bp were identified in *H. abietinum* and named Heterobasidion circular RNA 1 and 2 (HetcRNA 1 and 2). Both lack ORFs in their genomes (supporting their non-coding nature) and contain self-cleaving ribozymes, specifically, a hammerhead ribozyme (HHRz) and a deltavirus-like ribozyme (DVRz), one in each polarity, enabling symmetric rolling-circle replication. Predicted secondary structures (Figure 9) reveal a hybrid architecture, combining rod-like structures typical of ambiviruses with branched features that are more characteristic of plant viroids. These cRNAs were detected using the INFERNAL pipeline in the library containing *H. abietinum* RNAs (ABI). The pw identity between them was low enough (40.1%) to consider them as two different putative viroid-like cRNAs.

Recent metatranscriptomic studies have greatly expanded the known diversity of viroid-like circular RNAs, highlighting major challenges in their classification and even in defining the boundaries of this class of replicators [28,52]. Our findings align with a growing body of evidence suggesting that viroid-like elements may also occur in fungi [28,52,53,54]. Previous work has demonstrated viroid replication or the processing of its RNAs in the yeast *Saccharomyces cerevisiae* [55] and in the cyanobacterium *Nostoc* (*Nostocales*) [56]. Tian et al. [57] demonstrated that apple viroids can naturally spread to fungi, suggesting the possible existence of mycoviroids. Prior to these findings, Wei et al. [58] showed that plant viroids can infect and induce symptoms in fungal pathogens like *Cryphonectria parasitica* and *Fusarium graminearum*. In 2022, Afanasenko et al. [59] reported that the potato spindle tuber viroid (PSTVd) can persist in pure cultures of *Phytophthora infestans*, providing evidence of bidirectional viroid transfer between plants and oomycetes. In 2023, Botryosphaeria dothidea circular RNAs (BdcRNAs) were discovered in *Botryosphaeria dothidea*, showing viroid-like genomic features and pathogenic effects on fungal hosts [60].

Although the biological function and replication mechanisms of HetcRNA1 and HetcRNA2 remain to be experimentally verified, their genomic architecture strongly supports their classification as putative mycoviroid-like agents, contributing to the expanding landscape of non-coding circular RNAs associated with fungi. Definitive classification as true mycoviroids would require demonstration of replication by the fungal host’s RNA polymerase [28]. Without such evidence, it remains possible that replication is mediated by a co-infecting viral polymerase, in which case these elements would be more accurately described as mycoviroid-like satellites [61].

## 4. Conclusions

Our small-scale survey consisting of 17 Bosnian *H. annosum* s.l. specimens revealed the existence of 32 mycoviruses representing overall 11 dsRNA or ssRNA families and two putative novel mycoviroids (Table 1), suggesting that the high biodiversity of Bosnian forests is coupled with a similarly diverse viral community. For comparison, the screening of 45 Czech *H. annosum* s.l. strains using a similar NGS methodology detected 25 viruses belonging to nine ssRNA families [15]. A possible cause of the observed difference in virus abundance between the two countries could be that the authors of [15] surveyed managed forests, while the present work focused on near-natural stands, known to accommodate higher viral diversity than disturbed habitats [62,63]. Among dsRNA viruses, the presence of *O. annosi*, the most common mycovirus species in *Heterobasidion* spp. [14], was confirmed. New members of *Partitiviridae*, the family with the largest number of *Heterobasidion* viruses described to date, were also discovered. However, ssRNA viruses of *Heterobasidion*, especially ambi-like viruses, seem to be more prevalent in Bosnia and display a much higher genetic variability. We report the first beny-like virus and deltaflexivirus associated with the root rot pathogen. *Heterobasidion abietinum* is recorded as a host of ssRNA viruses (i.e., narna-, ourmia-, ambi-like, and mitoviruses) and putative mycoviroid-like elements for the first time. The identification of HetcRNAs expands upon this growing field, raising new questions about the functional roles and evolutionary significance of viroid-like RNAs in fungi and, in particular, forest fungal pathogens.

## Figures and Tables

**Figure 1 viruses-17-01144-f001:**
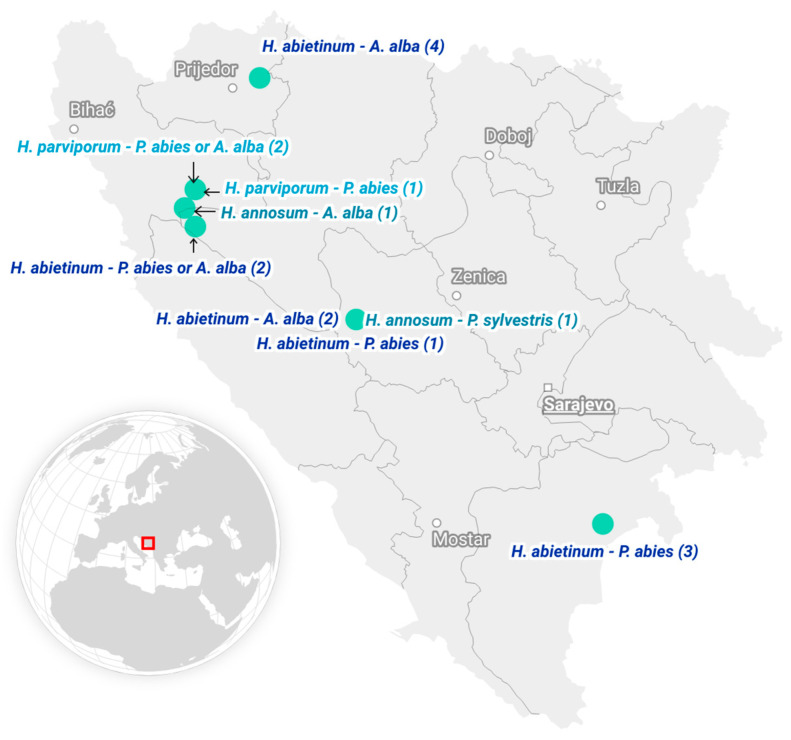
Map showing the sampling sites in Bosnia and Herzegovina together with the corresponding *Heterobasidion* spp. identified and the tree species from which they were collected. For complete information, see Appendix A. The map was created in www.datawrapper.de.

**Figure 2 viruses-17-01144-f002:**
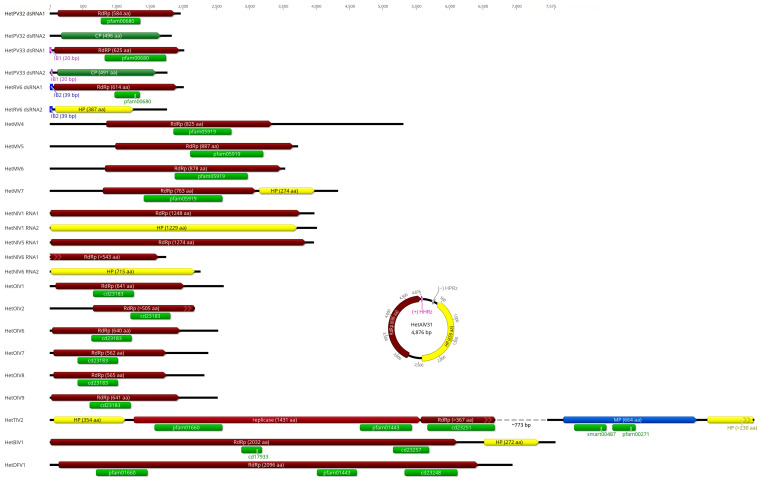
Genome organization of mycoviruses detected in *Heterobasidion*, drawn to scale (indicating genome length in nucleotides). One representative of the novel ambi-like viruses was selected to illustrate the typical location of ribozymes within their circular genome. HetPV, Heterobasidion partitivirus; HetRV, Heterobasidion RNA virus; HetMV, Heterobasidion mitovirus; HetNlV, Heterobasidion narna-like virus; HetOlV, Heterobasidion ourmia-like virus; HetTlV, Heterobasidion tobamo-like virus; HetBlV, Heterobasidion beny-like virus; HetDFV, Heterobasidion deltaflexivirus; HetAlV, Heterobasidion ambi-like virus. HPRz, hairpin ribozyme; HHRz, hammerhead ribozyme; IB, identical base. Predicted open reading frames: RdRP, RNA-dependent RNA polymerase; CP, coat protein; HP, hypothetical protein; MP, movement protein. Conserved domains: pfam00680, RdRP (E values: 1.7 × 10^−8^, 1.21 × 10^−12^, 4.53 × 10^−10^); pfam05919, mitovirus RdRP (1.25 × 10^−80^, 9.81 × 10^−88^, 2.99 × 10^−80^, 1.71 × 10^−88^); cd23183, *Botourmiaviridae* RdRP (3.92 × 10^−48^, 7.2 × 10^−47^, 2.82 × 10^−48^, 1.44 × 10^−52^, 7.12 × 10^−52^, 1.75 × 10^-51^); pfam01660, methyltransferase (1.3 × 10^−13^, 2.6 × 10^−24^); pfam01443, helicase (2.61 × 10^−29^, 3.39 × 10^−11^); cd23251, *Virgaviridae* RdRP (3.11 × 10^−54^); smart00487, DEAD-like helicases superfamily (1.47 × 10^−5^); pfam00271, helicase conserved C-terminal domain (1.44 × 10^5^); cd17933, DEXS-box helicase (6.52 × 10^−5^); cd23257, *Benyviridae* RdRP (4.0 × 10^−22^); cd23248, *Deltaflexiviridae* RdRP (2.22 × 10^−70^). The dashed line within the HetTlV2 genome symbolizes a gap, which could not be reconstructed from RNA-Seq data.

**Figure 4 viruses-17-01144-f004:**
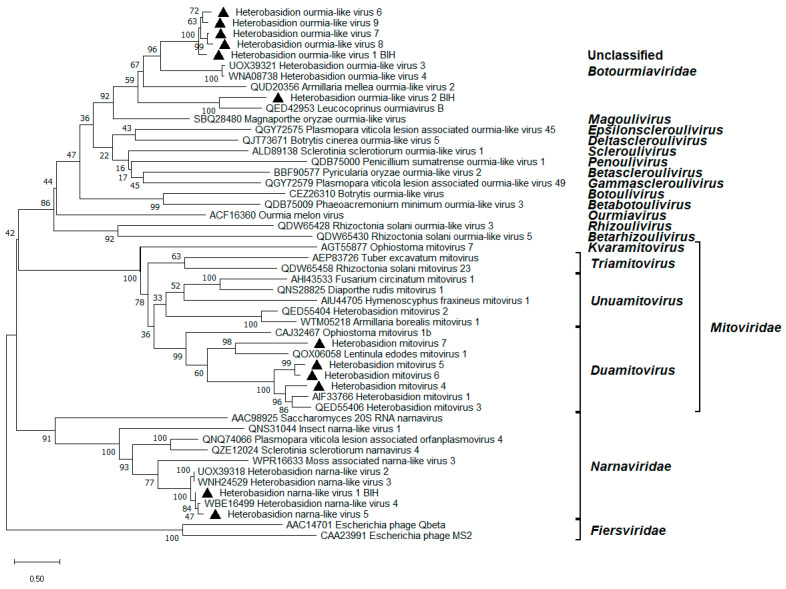
Phylogenetic relationships within and between relevant families of the *Lenarviricota* phylum. The *Fiersviridae* family served as outgroup. The tree was built based on the alignment of RdRP aa sequences generated with MUSCLE in MEGA11 [30]. The evolutionary history was inferred by using the Maximum Likelihood method and the Whelan And Goldman + Freq. model [45]. A discrete Gamma distribution was used to model evolutionary rate differences among sites (five categories +G +I). All positions with less than 95% site coverage were eliminated; i.e., fewer than 5% alignment gaps, missing data, and ambiguous bases were allowed at any position. Evolutionary analyses were conducted in MEGA11 with 1000 bootstrap repeats. The percentage of trees in which the associated taxa clustered together is shown next to the branches. Branch lengths are proportional to the number of substitutions per site. Ourmia-like, narna-like, and mitoviruses described in this study are denoted by a triangle.

**Figure 5 viruses-17-01144-f005:**
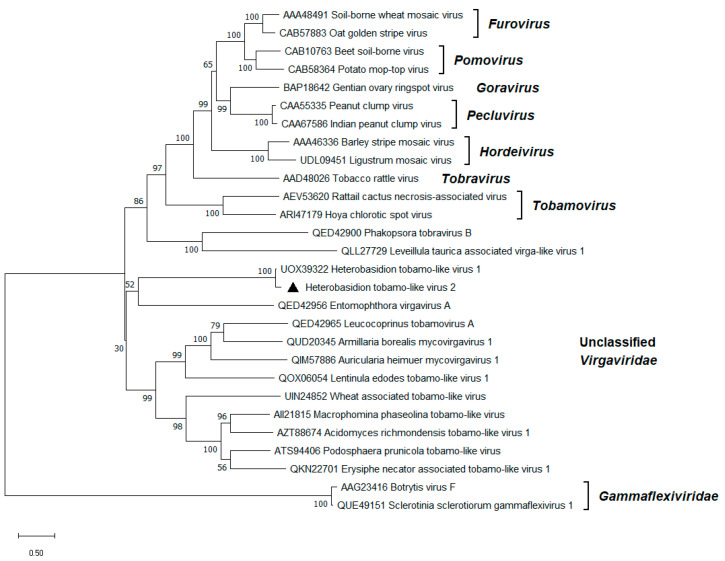
Phylogenetic relationships of HetTlV2 (denoted by a triangle) with selected members of the *Virgaviridae* family. The *Gammaflexiviridae* family served as outgroup. The tree was built based on the alignment of replication protein aa sequences generated with MUSCLE in MEGA11 [30]. The evolutionary history was inferred by using the Maximum Likelihood method and the Le_Gascuel_2008 model [41]. A discrete Gamma distribution was used to model evolutionary rate differences among sites (five categories +G +I). All positions with less than 95% site coverage were eliminated; i.e., fewer than 5% alignment gaps, missing data, and ambiguous bases were allowed at any position. Evolutionary analyses were conducted in MEGA11 with 1000 bootstrap repeats. The percentage of trees in which the associated taxa clustered together is shown next to the branches. Branch lengths are proportional to the number of substitutions per site.

**Figure 6 viruses-17-01144-f006:**
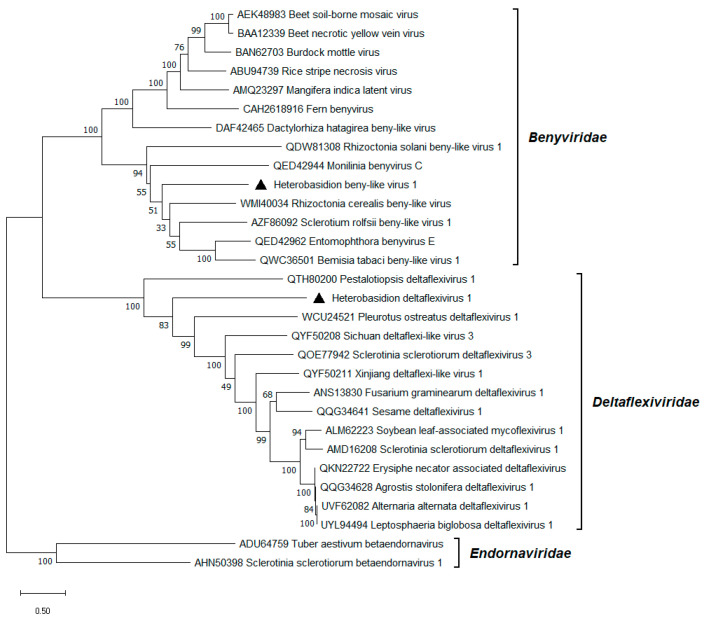
Phylogenetic relationships of HetBlV1 and HetDFV1 (both denoted by a triangle) with selected members of the *Benyviridae* and *Deltaflexiviridae* families. The *Endornaviridae* family served as outgroup. The tree was built based on the alignment of RdRP aa sequences generated with MUSCLE in MEGA11 [30]. The evolutionary history was inferred by using the Maximum Likelihood method and the Whelan And Goldman + Freq. model [45]. A discrete Gamma distribution was used to model evolutionary rate differences among sites (five categories +G +I). All positions with less than 95% site coverage were eliminated; i.e., fewer than 5% alignment gaps, missing data, and ambiguous bases were allowed at any position. Evolutionary analyses were conducted in MEGA11 with 1000 bootstrap repeats. The percentage of trees in which the associated taxa clustered together is shown next to the branches. Branch lengths are proportional to the number of substitutions per site.

**Figure 7 viruses-17-01144-f007:**
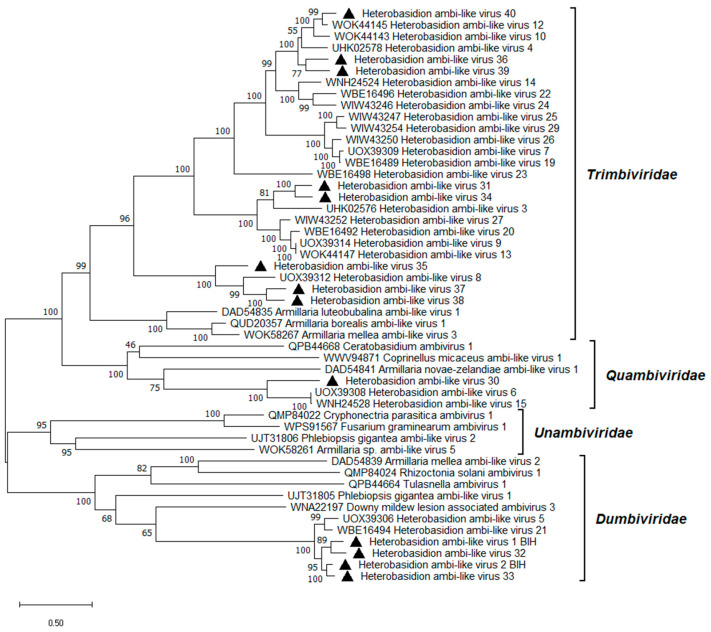
Phylogenetic relationships of ambi-like viruses. The tree was built based on the alignment of RdRP aa sequences generated with MUSCLE in MEGA11 [30]. The evolutionary history was inferred by using the Maximum Likelihood method and the Le_Gascuel_2008 model [41]. A discrete Gamma distribution was used to model evolutionary rate differences among sites (five categories +G +I). All positions with less than 95% site coverage were eliminated; i.e., fewer than 5% alignment gaps, missing data, and ambiguous bases were allowed at any position. Evolutionary analyses were conducted in MEGA11 with 1000 bootstrap repeats. The percentage of trees in which the associated taxa clustered together is shown next to the branches. Branch lengths are proportional to the number of substitutions per site. Viruses described in this study are denoted by a triangle.

**Figure 8 viruses-17-01144-f008:**
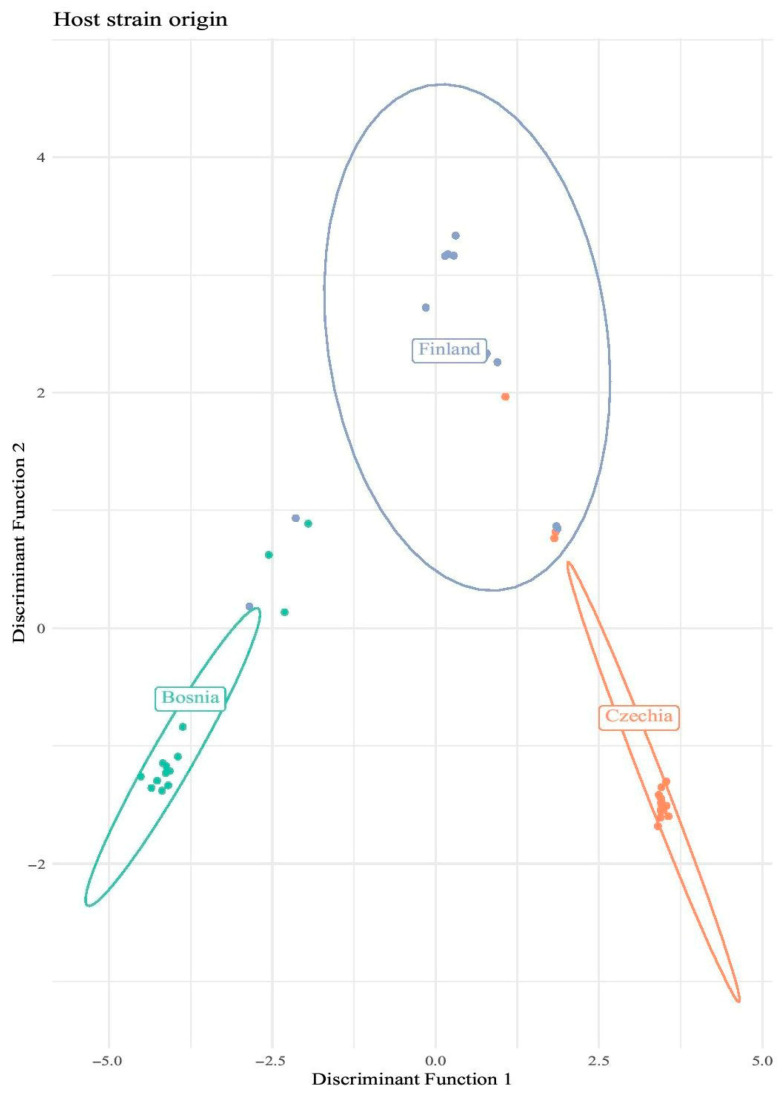
Country-wise DAPC plot of *Heterobasidion* ambiviruses. The two discriminant functions with the highest eigenvalues are displayed.

**Figure 9 viruses-17-01144-f009:**
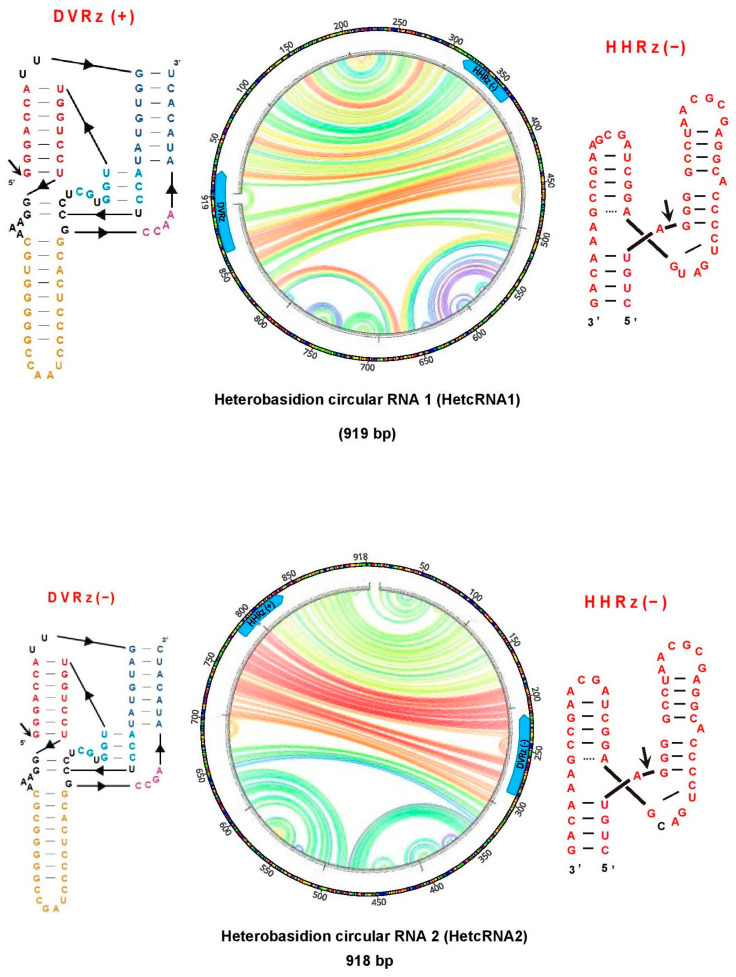
Illustration of the discovered HetcRNAs in the RNA library ABI. Viroid ribozymes. Circular plots of the predicted RNA secondary structures of both HetcRNAs.

## Data Availability

The RNA-Seq data have been deposited into the NCBI Sequence Read Archive database with the accession numbers SRR28491256, SRR28491257, SRR28491258, SRR28491259, SRR28496467, SRR28496590, and SRR28496591 under the BioProject PRJNA1093139. The mycoviral and mycoviroid-like genome sequences are available from the NCBI GenBank.

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
