# Peer review of "Diversity of RNA Viruses and Circular Viroid-like Elements in Heterobasidion spp. in Near-Natural Forests of Bosnia and Herzegovina"

_viruses, 2025, doi:10.3390/v17081144_

Round 1
Reviewer 1 Report
Comments and Suggestions for Authors
Manuscript viruses-3736338 describes the occurrence of viruses and small non-coding circular RNAs in Heterobasidion root rot fungi in forests of Bosnia and Herzegovina. The research is of interest and thoroughly conducted. In addition, the manuscript is very well written, although a few minor edits should be considered for improvement. See recommendations below:
Line 24: What does NGS stand for?
Line 28: ... virus families. In addition to the known circular ...
Line 31: ... is the first to report a putative ...
Lines 33-35: This sentence should be eliminated because other several reasons, including the use of less robust virus identification technologies, could explain the perceived higher diversity of Heterobasidion viruses in forests of Bosnia and Herzegovina.
Line 46: ... but also at the continental scale ...
Line 93: Eliminate Stranded
Line 97: Change sent to sequenced
Line 135: Spell out ORF
Line 145 and throughout the manuscript: Change RdRp to RdRP
Line 156: ... mycoviroid-like sequence segments ...
Line 182: Spell out CP
Author Response
Reviewer 1:
Manuscript viruses-3736338 describes the occurrence of viruses and small non-coding circular RNAs in Heterobasidion root rot fungi in forests of Bosnia and Herzegovina. The research is of interest and thoroughly conducted. In addition, the manuscript is very well written, although a few minor edits should be considered for improvement. See recommendations below:
Line 24: What does NGS stand for?
It means New Generation Sequencing. We have deleted NGS not to combine similar terms as NGS and RNA-Seq.
Line 28: ... virus families. In addition to the known circular ...
We have added the word “virus”
Line 31: ... is the first to report a putative ...
We have modified the sentence accordingly
Lines 33-35: This sentence should be eliminated because other several reasons, including the use of less robust virus identification technologies, could explain the perceived higher diversity of Heterobasidion viruses in forests of Bosnia and Herzegovina.
We have deleted this sentence as suggested by the reviewer.
Line 46: ... but also at the continental scale ...
We have modified the sentence accordingly
Line 93: Eliminate Stranded
We have deleted it as suggested by the reviewer
Line 97: Change sent to sequenced
We have modified the sentence
Line 135: Spell out ORF
We have spelled it out and added a short description
Line 145 and throughout the manuscript: Change RdRp to RdRP
We have done this change throughout the manuscript
Line 156: ... mycoviroid-like sequence segments ...
We have added this word throughout the manuscript
Line 182: Spell out CP
We have spelled it out
Reviewer 2 Report
Comments and Suggestions for Authors
The manuscript presents a brief report describing novel viruses associated with different species of the fungus Heterobasidion. The topic aligns with the journal's scope and may contribute relevant insights to the field. However, several important issues must be addressed before publication.
The viral sequences were obtained from a pooled sample of fungal isolates, and some of them exhibit atypical features when compared to classified viruses. These discrepancies may stem from bioinformatic artifacts, highlighting the need for additional validation. A comprehensive re-examination of the raw sequencing data, particularly SAM/BAM files, is strongly recommended to manually inspect read mappings and rule out potential assembly errors or chimeric sequences. Once these concerns are adequately addressed and any artifactual sequences are excluded, the manuscript should be re-evaluated to ensure the robustness and reliability of the findings.
Major comments:
Lines 31–33: The sentence "Our study is the first report of putative viroid-like circRNAs, a beny-like and a deltaflexivirus in Heterobasidion, and the first report of ssRNA viruses in Heterobasidion abietinum" should be revised. While the novelty is acknowledged, other RNA viruses without ICTV classification have been previously deposited in GenBank/NCBI. It is recommended that the authors rephrase this statement to reflect this context more accurately.
Lines 237–237: "HetMV7 also possessed a short downstream ORF encoding an HP (Figure 2) and earned the title of the longest mitovirus described to date ahead of HetMV3-an1 [42]. HetMV4 was represented by two virus strains, while HetMV7 by three (Table 1), with each strain sharing 98% aa pw identity over their RdRp." The authors refer to HetMV7 as the longest mitovirus described; however, Figure 2 appears to highlight HetMV4 instead. Moreover, the 3’-UTR of HetMV4 seems unusually long for a member of the Mitoviridae family. To determine whether this observation results from assembly artifacts or chimeric sequences, an analysis of the original FASTQ files and read mappings is essential. Mitoviruses typically encode a single protein, RNA-dependent RNA polymerase (RdRp). In light of the atypical length of HetMV4, a detailed reassessment of the sequencing data is strongly recommended to evaluate the integrity of the assembly and rule out technical artifacts.
Clarification on “Virus Strains”: The use of the term “virus strains” is unclear. It should be specified whether these sequences represent distinct contigs from a single fungal isolate or if they originate from different samples. Clarifying this point is necessary for the reader to assess the genetic diversity and biological relevance of the findings.
Taxonomic Classification: The authors appear to base taxonomic inference solely on BLASTx results. However, the classification of novel viruses should not rely exclusively on this approach. The use of pairwise sequence comparisons, such as PASC (Pairwise Sequence Comparison) or other alignment-based phylogenetic methods, is essential to ensure proper taxonomic placement.
Lines 397–398: "Two small circular RNAs (circRNAs) of 918 and 919 bp were recently identified in H. abietinum and named Heterobasidion circular RNA 1 and 2 (HetcRNA 1 and 2)." It is unclear whether these sequences were identified in the current study or had been previously reported. A clarifying statement and appropriate citation, if applicable, would enhance the transparency and reproducibility of the manuscript. Additionally, viroids typically range from 200 to 500 nucleotides in length. The described sequences, being considerably larger, do not correspond to known "mycoviroids." The secondary structure analysis presented is insufficient, and the identification of a hammerhead ribozyme alone is not sufficient for classification as a viroid. Given the sequence length, the presence of potential ORFs should be investigated and followed by homology searches in public protein databases to explore the nature of these elements further.
Recommendation:
At this stage, I recommend that the manuscript not be accepted for publication in its current form. However, I encourage the authors to address the points raised above. Upon careful revision and validation of the data, the study may be reconsidered for publication in a subsequent round of review.
Author Response
Reviewer 2:
The manuscript presents a brief report describing novel viruses associated with different species of the fungus Heterobasidion. The topic aligns with the journal's scope and may contribute relevant insights to the field. However, several important issues must be addressed before publication.
The viral sequences were obtained from a pooled sample of fungal isolates, and some of them exhibit atypical features when compared to classified viruses. These discrepancies may stem from bioinformatic artifacts, highlighting the need for additional validation. A comprehensive re-examination of the raw sequencing data, particularly SAM/BAM files, is strongly recommended to manually inspect read mappings and rule out potential assembly errors or chimeric sequences. Once these concerns are adequately addressed and any artifactual sequences are excluded, the manuscript should be re-evaluated to ensure the robustness and reliability of the findings.
We appreciate this comment and following the reviewer’s suggestion and to have a more rigorous explanation of our results we have added the coverage plots and bam files with the reads mapped to every virus contig to a folder which can be accessed in this link: 10.6084/m9.figshare.29634521. We have added this information to the manuscript.
Major comments:
Lines 31–33: The sentence "Our study is the first report of putative viroid-like circRNAs, a beny-like and a deltaflexivirus in Heterobasidion, and the first report of ssRNA viruses in Heterobasidion abietinum" should be revised. While the novelty is acknowledged, other RNA viruses without ICTV classification have been previously deposited in GenBank/NCBI. It is recommended that the authors rephrase this statement to reflect this context more accurately.
We understand reviewer’s point and we have modified our statement. We wrote: “This study documents the first report of a putative viroid-like RNA agent in Heterobasidion, along with beny-like and deltaflexivirus-like viruses in Heterobasidion abietinum, and expands the known virosphere of Heterobasidion species in Southeastern European forests”
Lines 237–237: "HetMV7 also possessed a short downstream ORF encoding an HP (Figure 2) and earned the title of the longest mitovirus described to date ahead of HetMV3-an1 [42]. HetMV4 was represented by two virus strains, while HetMV7 by three (Table 1), with each strain sharing 98% aa pw identity over their RdRp." The authors refer to HetMV7 as the longest mitovirus described; however, Figure 2 appears to highlight HetMV4 instead. Moreover, the 3’-UTR of HetMV4 seems unusually long for a member of the Mitoviridae family. To determine whether this observation results from assembly artifacts or chimeric sequences, an analysis of the original FASTQ files and read mappings is essential. Mitoviruses typically encode a single protein, RNA-dependent RNA polymerase (RdRp). In light of the atypical length of HetMV4, a detailed reassessment of the sequencing data is strongly recommended to evaluate the integrity of the assembly and rule out technical artefacts.
We thank the reviewer for their careful observation. The reference to HetMV7 as the longest mitovirus described was indeed incorrect; it should have referred to HetMV4, which we have now corrected in the text. We also acknowledge the reviewer’s concern regarding the unusually long 3′-UTR of HetMV4.
To address the possibility of assembly artefacts or chimeric sequences, we conducted a thorough examination of the original FASTQ files and read mappings. As detailed in the revised manuscript, we have now included coverage plots generated in Geneious for all viral contigs, including mitoviruses, and made the corresponding BAM files available via FigShare (DOI: 10.6084/m9.figshare.29634521). These files demonstrate consistent and high coverage across the full length of each viral contig, including the 3′-UTR of HetMV4. Also, we would like to point that Table 1 has the detailed information on read depth and total read counts for each virus. In the case of the tobamo-like virus, which has the lowest coverage depth, a short discussion is included.
Based on this analysis, we found no evidence of assembly artefacts or chimeric regions, and we have clarified this point in the Results section. We are confident that the HetMV4 and 7 sequence represents a genuine, full-length viral genome.
Clarification on “Virus Strains”: The use of the term “virus strains” is unclear. It should be specified whether these sequences represent distinct contigs from a single fungal isolate or if they originate from different samples. Clarifying this point is necessary for the reader to assess the genetic diversity and biological relevance of the findings.
We agree with the reviewer and have changed the term “strain” for “variant, which is more correct in this case because we have not studied the possible distinct biological properties of these putative viruses.
Taxonomic Classification: The authors appear to base taxonomic inference solely on BLASTx results. However, the classification of novel viruses should not rely exclusively on this approach. The use of pairwise sequence comparisons, such as PASC (Pairwise Sequence Comparison) or other alignment-based phylogenetic methods, is essential to ensure proper taxonomic placement.
We thank the reviewer for this important observation. We agree that BLASTx alone is insufficient for accurate taxonomic classification of novel viruses. In our study, we considered not only BLASTx results but also phylogenetic analyses and pairwise sequence comparisons among to distinguish viruses and their possible variants. While we did not specifically refer to the method as PASC, we performed standard pairwise (pw) comparisons, and we hope this is clearly presented in the Results section. To improve clarity, we have now explicitly described this approach in the revised version of the Materials and Methods.
Lines 397–398: "Two small circular RNAs (circRNAs) of 918 and 919 bp were recently identified in H. abietinum and named Heterobasidion circular RNA 1 and 2 (HetcRNA 1 and 2)." It is unclear whether these sequences were identified in the current study or had been previously reported. A clarifying statement and appropriate citation, if applicable, would enhance the transparency and reproducibility of the manuscript.
We have deleted the word “recently” because it was mistakenly place in the text. The putative viroid-like elements are reported in this manuscript for the first time.
Additionally, viroids typically range from 200 to 500 nucleotides in length. The described sequences, being considerably larger, do not correspond to known "mycoviroids." The secondary structure analysis presented is insufficient, and the identification of a hammerhead ribozyme alone is not sufficient for classification as a viroid. Given the sequence length, the presence of potential ORFs should be investigated and followed by homology searches in public protein databases to explore the nature of these elements further.
We thank the reviewer for this valuable observation. Indeed, the classical definition of viroids encompasses small, non-coding circular RNAs of 246–401 nt, exclusively infecting plants. However, recent metatranscriptomic studies (e.g., Forgia et al., Nat. Commun. 2023; Lee et al., Cell 2023) have revealed the existence of a broad diversity of circular RNAs—now referred to as viroid-like agents—that exhibit hallmark features of viroids, including circularity, stable secondary structure, ribozymes, but often fall outside the canonical size range and may contain putative ORFs. Many of these newly identified RNAs range from 500 to over 2000 nt and are found in diverse hosts, including fungi. In our study, we have been careful to refer to these elements as putative mycoviroid-like agents and viroid-like, not as bona fide viroids. But, we have clarified in both the main text and the conclusions that these are putative novel mycoviroid-like agents, and not formally classified viroids. Demonstration of replication by the fungal host’s RNA polymerase would be required for their classification as true mycoviroids—an important point which we now also emphasize in the Discussion (page 8).
Following the reviewer’s suggestion, we have re-examined the sequences for potential ORFs and performed homology searches in public protein databases (NCBI nr and Pfam). No significant ORF homologues were detected, supporting the probability that these are non-coding elements.

Reviewer 3 Report
Comments and Suggestions for Authors
This study investigated the RNA viruses and viroid-like genomes of Heterobasidion root rot fungi in near-natural forests of Bosnia and Herzegovina (Dinaric Alps), marking the first such research in this region. Through RNA-Seq and bioinformatic analyses of 17 isolates of Heterobasidion annosum sensu lato, a total of 32 mycoviruses were discovered, 26 of which were novel. These viruses include dsRNA viruses, linear ssRNA viruses, and circular ssRNA viruses. Additionally, two new non-coding circRNAs were identified, encoding HHRz and deltavirus ribozymes, respectively. This study is the first to report the presence of putative viroid-like circRNAs, a beny-like, and a deltaflexivirus in Heterobasidion, as well as ssRNA viruses in Heterobasidion abietinum. The results indicate that the high biodiversity of Bosnian forests is closely associated with a rich viral community. However, the manuscript still has some shortcomings at present. Therefore, I suggest making major revisions to the manuscript. Some comments are provided below.
- Although the study cites a large number of references, some of the references may be outdated or may not cover the latest research findings. This could affect the cutting-edge nature and scientific validity of the study.
- The sample size is relatively small (only 17 strains) and mainly from the near-natural forests of Bosnia and Herzegovina. This limited sample size may not fully reflect the diversity of the Heterobasidion viral community, especially across a broader geographic area and ecosystem.
- The study mainly focuses on the discovery and classification of viruses, and there is a lack of experimental data to prove whether these viruses can really reduce the pathogenicity of Heterobasidion. There is a lack of validation of the biological functions of these viruses.
- The study primarily focuses on Heterobasidion, but does not explore whether these viruses can infect other related fungal species. This may limit the understanding of the viral niches and their dissemination range.
- Although the study has discovered many novel viruses, it does not explicitly indicate how these findings can be translated into practical biocontrol strategies.
- Although a variety of bioinformatics tools were used, the limitations of these tools may lead to the omission or misclassification of certain viruses. Cross-validation using multiple different bioinformatics tools can reduce the possibility of misjudgment.
Author Response
Reviewer 3
This study investigated the RNA viruses and viroid-like genomes of Heterobasidion root rot fungi in near-natural forests of Bosnia and Herzegovina (Dinaric Alps), marking the first such research in this region. Through RNA-Seq and bioinformatic analyses of 17 isolates of Heterobasidion annosum sensu lato, a total of 32 mycoviruses were discovered, 26 of which were novel. These viruses include dsRNA viruses, linear ssRNA viruses, and circular ssRNA viruses. Additionally, two new non-coding circRNAs were identified, encoding HHRz and deltavirus ribozymes, respectively. This study is the first to report the presence of putative viroid-like circRNAs, a beny-like, and a deltaflexivirus in Heterobasidion, as well as ssRNA viruses in Heterobasidion abietinum. The results indicate that the high biodiversity of Bosnian forests is closely associated with a rich viral community. However, the manuscript still has some shortcomings at present. Therefore, I suggest making major revisions to the manuscript. Some comments are provided below.
Although the study cites a large number of references, some of the references may be outdated or may not cover the latest research findings. This could affect the cutting-edge nature and scientific validity of the study.
We appreciate the reviewer’s concern regarding the timeliness of the references. Although some of the references are not from the most recent years, they represent highly relevant work in the field that remains scientifically valid. Where appropriate, we have cited recent literature, including studies published after 2022, especially regarding the viruses and viroid-like elements that have recently emerged. We therefore believe the citation strategy used supports both the scientific validity and the relevance of the manuscript. We have added a couple of recent references that we could have missed in the previous version of the manuscript.
The sample size is relatively small (only 17 strains) and mainly from the near-natural forests of Bosnia and Herzegovina. This limited sample size may not fully reflect the diversity of the Heterobasidion viral community, especially across a broader geographic area and ecosystem.
We agree with the reviewer that including a larger number of Heterobasidion isolates would provide a more comprehensive view of the virome in Bosnia and Herzegovina. However, the sampling was constrained by the time-limited research stay of László Dalya in Banja Luka under the Erasmus+ program. Despite this limitation, we believe that the isolates analyzed in this study offer a valuable first vision into the Heterobasidion-associated virome in this region and lay the groundwork for future, broader surveys.
The study mainly focuses on the discovery and classification of viruses, and there is a lack of experimental data to prove whether these viruses can really reduce the pathogenicity of Heterobasidion. There is a lack of validation of the biological functions of these viruses.
We appreciate the reviewer’s comment and agree that experimental validation of the biological effects of the detected viruses on Heterobasidion pathogenicity would provide important functional insights. However, the main objective of this study was to characterize the viral diversity associated with Heterobasidion in a previously unexplored geographic region.
The study primarily focuses on Heterobasidion, but does not explore whether these viruses can infect other related fungal species. This may limit the understanding of the viral niches and their dissemination range.
We thank the reviewer for this suggestion. While we agree that investigating the potential host range of the identified viruses could provide valuable ecological insights, this study was intentionally focused on Heterobasidion, in line with the goals of Laszlo Dalya’s PhD research. Exploring cross-species infectivity would require a broader sampling strategy and experimental assays, which were beyond the scope of this study.
Although the study has discovered many novel viruses, it does not explicitly indicate how these findings can be translated into practical biocontrol strategies.
We appreciate the reviewer’s comment regarding the potential for biocontrol applications. While the current study focuses on the discovery and characterization of novel viruses associated with Heterobasidion, time constraints during the fieldwork and sequencing phases did not allow us to experimentally assess their biological effects. However, we have addressed this aspect in a separate study (Dalya et al. 2024), in which we experimentally investigated the impact of selected viruses on Heterobasidion growth and pathogenicity.
Although a variety of bioinformatics tools were used, the limitations of these tools may lead to the omission or misclassification of certain viruses. Cross-validation using multiple different bioinformatics tools can reduce the possibility of misjudgment.
We agree that all bioinformatic approaches have limitations, particularly when identifying highly divergent or novel viral sequences. However, the pipelines used in this study are robust, have been carefully optimized, and have been successfully applied in several of our team’s recent publications. To decrease the risk of omission or misclassification, we combined multiple complementary approaches, including sequence similarity searches and conserved domain detection. We have clarified this in the Methods section to better reflect the reliability of our virus discovery strategy.

Round 2
Reviewer 3 Report
Comments and Suggestions for Authors
I recommend the acceptance of the revised manuscript for publication.